# The PANDEMYC Score. An Easily Applicable and Interpretable Model for Predicting Mortality Associated With COVID-19

**DOI:** 10.3390/jcm9103066

**Published:** 2020-09-23

**Authors:** Juan Torres-Macho, Pablo Ryan, Jorge Valencia, Mario Pérez-Butragueño, Eva Jiménez, Mario Fontán-Vela, Elsa Izquierdo-García, Inés Fernandez-Jimenez, Elena Álvaro-Alonso, Andrea Lazaro, Marta Alvarado, Helena Notario, Salvador Resino, Daniel Velez-Serrano, Alejandro Meca

**Affiliations:** 1University Hospital Infanta Leonor, 28031 Madrid, Spain; juan.torresm@salud.madrid.org (J.T.-M.); jorge_vlr@yahoo.es (J.V.); marioperezb@hotmail.com (M.P.-B.); ejgonzalezbuitrago@salud.madrid.org (E.J.); mario.fontan@salud.madrid.org (M.F.-V.); elsa.izquierdo@salud.madrid.org (E.I.-G.); ifernandezj@salud.madrid.org (I.F.-J.); elenaalba.alvaro@salud.madrid.org (E.Á.-A.); andrea.lazaro@salud.madrid.org (A.L.); marta.alvarado@salud.madrid.org (M.A.); nelena.notario@gmail.com (H.N.); 2Department of Mathematics, Complutense de Madrid University (UCM), 28040 Madrid, Spain; danielvelezserrano@mat.ucm.es; 3Instituto de Investigación Sanitaria Gregorio Marañón (IiSGM), 28007 Madrid, Spain; 4Unidad de Infección Viral e Inmunidad, Instituto de Salud Carlos III, 28007 Madrid, Spain; Salvador.resino@gmail.com; 5Department of Preventive Medicine & Public Health, Rey Juan Carlos University, 28933 Madrid, Spain; alalvaromeca@gmail.com

**Keywords:** SARS-CoV-2, COVID-19, prediction score, mortality

## Abstract

This study aimed to build an easily applicable prognostic model based on routine clinical, radiological, and laboratory data available at admission, to predict mortality in coronavirus 19 disease (COVID-19) hospitalized patients. Methods: We retrospectively collected clinical information from 1968 patients admitted to a hospital. We built a predictive score based on a logistic regression model in which explicative variables were discretized using classification trees that facilitated the identification of the optimal sections in order to predict inpatient mortality in patients admitted with COVID-19. These sections were translated into a score indicating the probability of a patient’s death, thus making the results easy to interpret. Results. Median age was 67 years, 1104 patients (56.4%) were male, and 325 (16.5%) died during hospitalization. Our final model identified nine key features: age, oxygen saturation, smoking, serum creatinine, lymphocytes, hemoglobin, platelets, C-reactive protein, and sodium at admission. The discrimination of the model was excellent in the training, validation, and test samples (AUC: 0.865, 0.808, and 0.883, respectively). We constructed a prognostic scale to determine the probability of death associated with each score. Conclusions: We designed an easily applicable predictive model for early identification of patients at high risk of death due to COVID-19 during hospitalization.

## 1. Introduction

Despite substantial efforts to prevent the spread of coronavirus 19 disease (COVID-19), at the end of June 2020 over 14 million people worldwide had tested positive for SARS-CoV-2, and more than 603,000 had died [1].

During March and April 2020, Spain had one of the highest rates of COVID-19 and had experienced one of the most severe outbreaks of the disease worldwide. The rate of infections in the Autonomous Region of Madrid has exceeded that of every other region in Spain, with more than 28% of all confirmed cases and a cumulative total of 42,747 hospitalized patients and 8441 deaths [2,3]. The excessive workload generated by the COVID-19 pandemic has led to drug shortages and an insufficient number of conventional and intensive care beds.

The wide variation in the symptoms of COVID-19 makes it difficult to predict the clinical course, thus complicating triage. Clinical experience has demonstrated significant heterogeneity in the course of severe acute respiratory syndrome coronavirus 2 (SARS-CoV-2) infection: while some patients are asymptomatic or progress with mild symptoms, others develop severe acute respiratory distress syndrome with multiorgan failure and death [4]. In addition, it is very difficult to accurately predict clinical outcomes in patients with such a myriad of clinical presentations.

Early detection of cases that are at high risk of progression to severe COVID-19 with an imminent risk of death and identification of contributing factors are now urgent and challenging. Accurate prediction of the mortality of COVID-19 would therefore enable targeted strategies that facilitate appropriate and early supportive care and enable patients to be categorized according to severity and prognosis. This is especially important in outbreaks, where prioritization of patients can reduce unnecessary or inappropriate use of health care resources [5].

We designed a reliable prognostic model using routine, widely available demographic, clinical, and laboratory data to predict mortality in patients hospitalized with COVID-19.

## 2. Experimental Section

### 2.1. Study Design and Data Source

We performed a retrospective observational study at Infanta Leonor University Hospital (ILUH), a secondary level hospital with 361 beds (including eight intensive care beds) in Vallecas, an area in the southeast of Madrid that is home to more than 305,000 inhabitants. Vallecas was one of the areas most affected by COVID-19 in the city of Madrid, with 4713 total confirmed cases as of July 7th, 2020 [3]. Therefore, the level of hospital saturation during the pandemic outbreak of COVID-19 was one of the highest in Spain. Consequently, in March, the hospital became a COVID-19 center, and all its healthcare professionals focused solely on infected patients.

The study population comprised all patients admitted to hospital with a confirmed diagnosis of COVID-19 based on a positive result in the SARS-CoV-2 reverse transcriptase-polymerase chain reaction assay between 2nd March and 31st May 2020. Samples were obtained via nasopharyngeal swabs.

In order to construct our predictive score, we collected demographic data, previous diseases, clinical presentation, and laboratory and radiological data at admission from electronic medical records and managed our findings using REDCap (Research Electronic Data Capture, Vanderbilt University, 2201 West End Ave, Nashville, TN 37235, USA) hosted at the Ideas for Health Association. REDCap is a secure, web-based software platform designed to support data capture for research studies [6].

### 2.2. Ethical Aspects

The study was approved by the Institutional Investigation and Ethics Review Board of Infanta Leonor University Hospital (CEI-ILUH) (Code ILUH R 027-20). Given the retrospective nature of the study, the need for informed consent from patients was waived. Data confidentiality was maintained at all times according to Spanish legislation.

### 2.3. COVID-19 Cohort Identification

The primary cohort was divided into a training, a validation, and a test sample, with 40%, 30%, and 30% of the observations reserved, respectively, for each.

### 2.4. Model Development and Statistical Analysis

In order to fit a logistic regression model, data were previously processed to avoid the effect of outliers, include missing values, and to consider possible non-linear associations between the target and the input variables.

During this preprocessing step we discretized the continuous variables into a specific number of categories and then generated a binary or dummy variable for each one. Discretization was performed using classification trees, one for each input. The aim was to identify cut-off points in the observed input values that discriminate as much as possible between the different target categories in order to enhance the predictive capacity of the transformed variables.

By definition, missing values and outliers are not present in binary variables. Moreover, the estimation of a value for each of these variables makes it possible to consider non-linear effects. The disadvantage of this approach is that a huge number of dummy variables can be generated. We solved this problem by applying a data transformation method that is widely used in the credit scoring field [7].

The method generates a new continuous variable (WoEx) for each input variable X. The values are obtained by averaging the target in the different categories obtained through the discretization process. This transformation procedure is usually known as the WoE (Weight of Evidence) transformation [8]. WoE measures the strength of an input variable for differentiating between the classes of the target variable. In this study, WoE measures the proportion of dead patients to live patients at each group level.

With Y as a binary dependent variable, X as an explanatory variable, and Xc as the associated intermediate variable obtained after discretizing X into the XI attributes x1, xI, the value of WX at each category xi is defined as:(1)$$W_{X}\left(x_{i}\right)=ln\left(\frac{P\left(Y=1|X_{C}=x_{i}\right)}{P\left(Y=0X_{C}=x_{i}\right)}\right)=ln\left(ODD\left(x_{i}\right)\right)$$

This value is derived by taking the logarithm transformation over the odds values associated with each of the categories *x_i_* derived from the original input variable.

WoE methodology overcomes previous selection problems and prevents the creation of an elevated number of dummy variables. Indeed, this approach generates only one transformed variable per input and, therefore, only one *p*-value when deciding whether or not to keep the variable in the regression model.

If a WoE variable, W_X_, is included in the regression model, the odds ratios cannot be used to explain the effect of the original variable X over the dependent variable, because W_X_ is not measured in the same units as X. Therefore, a linear transformation of the value resulting from the multiplication of $W_{X}\left(x_{i}\right)$ and the regression coefficient estimated for W_X_ is carried out. The aim of this transformation is to scale this value to a particular range of scores [9].

The score points are proportional to the logarithm of the predicted death/non-death odds of the patient. For each attribute $x_{i}$ of the variable, X is calculated as follows:(2)$$SCORE_{X}\left(x_{i}\right)=\left(-W_{X}\left(x_{i}\right)*\beta +\frac{\alpha }{n}\right)*factor+\frac{offset}{n}$$
where

$W_{X}\left(x_{i}\right)$ is the weight of evidence of the ith attribute of the characteristic X

β is the regression coefficient associated with the variable X

α is the intercept of the logistic regression model

*n* is the number of variables in the logistic regression model

Factors and offset values are scaling parameters that enable the analyst to control the range of the scores and the rate of change in odds for a given increase in the score. The sum of the scores associated with each of the input variables provides a total score for the patient.

### 2.5. Calibration

For purposes of calibration, patients in the test database were split into deciles ordered by their probability of death. For each decile, the mean of the predicted probability of death was calculated and compared with the mean observed probability of death.

Moreover, to evaluate predictive capacity, the model was compared with other competitive machine learning methods. Based on the method described by DeLong et al. [10], these results were measured using the area under the curve (AUC) in the data test. The model that yielded the best results was the gradient boosting type. However, these are not significantly better in statistical terms than those obtained with the WoE methodology. In addition, those of the latter seem more reliable when generalizing to other samples, since the difference in the AUC obtained with the training and test tables is significantly lower than in the case of gradient boosting (Appendix A). Considering these observations and the difficulty in interpreting the results obtained with gradient boosting, we have chosen to consider the WoE methodology as the most appropriate to address a problem of this type.

The results are presented as the median (interquartile range) for continuous variables and as frequencies and percentages for categorical variables. Categorical data and proportions were analyzed using the chi-squared test or Fisher’s exact test, as required. The Mann-Whitney test was used to compare continuous variables. In the case of multiple comparisons, a Bonferroni correction was applied to compare the results between the groups (α = 0.05). The statistical analysis was performed using the statistical package R, foundation for Statistical Computing, Vienna, Austria. URL http://www.R-project.org/ version 3.1.1 (GNU General Public License) and SAS Enterprise Miner (SAS Institute, Cary NC, USA). All tests were two-tailed, and *p*-values < 0.05 were considered significant.

## 3. Results

### 3.1. Patient Cohorts

The study population comprised 1968 patients. The baseline patient characteristics were as follows: median age was 67 (27) years, and 1104 patients (56.4%) were male. A total of 325 patients died during hospitalization (17.7%). Hypertension was the most frequent comorbidity (991 patients (51.8%)), followed by dyslipidemia (36.8%), smoking (28%), and chronic heart disease (21.7%). The most frequent symptoms at admission were fever (75.1%), cough (66.7%), and dyspnea (56.8%).

Median oxygen saturation at admission was 94% (Interquartile Range, IQR= 7) (Table 1).

Patients who died during hospitalization were older (median age, 82 vs. 63 years; *p* < 0.001) and mainly male (68.7 vs. 54%; *p* < 0.001), with more comorbidities, such as hypertension, dyslipidemia, smoking/ex-smoking, and chronic heart disease. Dyspnea, cough, myalgia, headache, and low level of consciousness were more frequent in patients who died. These patients also had lower oxygen saturation (89 (11) vs. 95 (5); *p* < 0.001).

When the cohort was divided into derivation and validation cohorts, we did not find significant differences in the distribution of the variables included in the model (Appendix A).

### 3.2. Model Development

The final model included the following variables: age; oxygen saturation at admission; smoking or ex-smoking; and serum creatinine, lymphocytes, hemoglobin, sodium, platelets, and C-reactive protein at admission. We used these variables to build the acronym PANDEMYC (Platelets, Age, Natremia, Kidney injury, Lymphopenia, Oxygen saturation, C-reactive protein) to name the score (Figure 1)

### 3.3. Validation

To evaluate predictive capacity, the model was compared with other competitive machine learning methods (Appendix A). The Transparent Reporting of a multivariable prediction model for Individual Prognosis or Diagnosis (TRIPOD) checklist was used to ensure the quality of our model (Appendix A).

Our model showed good predictive capacity in the training, validation, and test samples, with an AUC of 0.865, 0.808, and 0.883, respectively. However, the presence of high AUC values did not mean that the individual probability of each class was well calibrated. Appendix A shows our probability calibration according to test data, which is good owing to the location of all points on the ideal line. When we compared our model with other models obtained using highly competitive machine learning methods such as gradient boosting, we did not find statistically significant differences (Appendix A).

## 4. Discussion

Several studies have reported risk factors associated with death in patients with COVID-19, although very few propose reliable prediction models, which should be constructed using an adequate sample size and a standardized methodology to avoid significant bias [5,11,12,13,14].

PANDEMYC showed high diagnostic accuracy (AUC = 0.88), as reported elsewhere [15], revealed high-risk factors, and established a simple, intuitive, and quantitative method for accurately estimating the risk of death. The score is based on variables that are quickly and easily collected and reproducible in any hospital at admission (web page calculator www.pandemyc-score.com). This simple model enables us to prioritize patients, especially during a pandemic, when limited healthcare resources have to be allocated. It also makes it possible to select patients for early discharge or to establish more intensive monitoring and treatment [16]. For example, patients with a score < 200 points and no significant respiratory insufficiency could be evaluated for early discharge. In our cohort, only two of 420 patients (0.4%) with these characteristics died during hospitalization.

All the factors included were clinically significant, with age, oxygen saturation, and serum creatinine at admission being the most relevant variables according to their weight and significance in the regression model. This observation is consistent with previous reports, thus showing the importance of aging, respiratory distress, and kidney injury as prognostic factors [12,14].

Analytical variables such as C-reactive protein, lymphopenia, and anemia indicate the association between the degree of inflammation and the risk for severe COVID-19 [15]. Thrombopenia is probably related to imbalance of the coagulation system and systemic thrombotic microangiopathy, which have been described in COVID-19 patients and are closely associated with mortality [17]. Finally, smoking is also a prognostic factor and may be associated with endothelial damage, as may other factors [18]. We did not detect significant values for variables that were frequently significant in previous models, such as gender and hypertension.

Despite the nature of the target classification variable, two main approaches are commonly distinguished for addressing classification problems, namely, interpretable methods and predictive methods [19]. The former typically place more emphasis on ensuring that the models obtained are easy to interpret, whereas the latter focus more on predictive accuracy. Consequently, the approaches are considered, at least to some extent, mutually exclusive, i.e., interpretability or predictive accuracy comes at the price of the other. We believe that interpretability is a key characteristic of any predictive model for a new disease, albeit with considerable caveats.

It is noteworthy that our model assigns a score to the missing values of each variable, thus creating a realistic model for real-life practice. Our model is not influenced by outliers, because they are not affected in the discretization process and therefore play no role in score assignment.

The main limitation of this study is that it was performed in a single center with no external validation. Nevertheless, the predictive capacity of the model was assured, because the AUC between the training and test samples, without the participation of the last in the adjustment process of the model, were very similar (0.865 and 0.883 respectively). Another limitation is that the level of saturation could have affected patient outcome. However, as saturation involves many components (available ICU beds, patient/physician ratio, new admissions, number of transfers to other hospital on a daily basis), we were unable to investigate it.

## 5. Conclusions

In conclusion, we designed a reliable, easily applicable prognostic score that can be applied in limited-resource settings to optimize management of patients hospitalize with COVID-19.

## Figures and Tables

**Figure 1 jcm-09-03066-f001:**
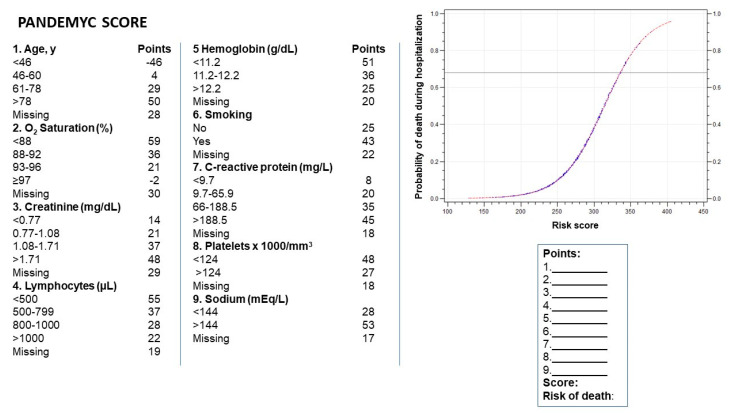
Resulting score for each variable during the discretization process. The mortality risk is calculated by summing the points for each variable.

**Table 1 jcm-09-03066-t001:** Demographics and comorbidity data of 1968 hospitalized patients with COVID-19 stratified by survival.

Characteristics	Alive *n* = 1643	Dead *n* = 325	*p*-Value
Male sex, No. (%)	882 (54)	222 (68.7)	<0.001
Age, Median (IQR)	63 (26)	82 (16)	<0.001
Born in Spain, No. (%)	1132 (72.8)	281 (89.2)	<0.001
Dead, No. (%)	0 (0)	325 (100)	<0.001
Comorbid Conditions, No. (%)			
Chronic Heart Disease	283 (17.8)	132 (41.2)	<0.001
Hypertension	765 (48)	226 (70.6)	<0.001
Chronic Pulmonary Disease	169 (10.7)	69 (21.8)	<0.001
Asthma	138 (8.7)	18 (5.6)	0.103
Stage 4 Chronic Kidney Disease	68 (4.3)	42 (13.2)	<0.001
Liver Cirrhosis	22 (1.4)	9 (2.8)	0.099
Solid Neoplasm (Active)	42 (2.6)	39 (12.3)	<0.001
Hematologic Neoplasm (Active)	21 (1.3)	13 (4)	0.001
HIV Infection	11 (0.7)	0 (0)	0.284
Obesity	239 (17.8)	45 (17.3)	0.809
Diabetes	337 (21.2)	99 (31)	<0.001
Dyslipidemia	347 (33.8)	97 (53.6)	0.001
Inflammatory Disease	74 (4.7)	25 (7.8)	0.024
Dementia	71 (4.5)	37 (11.7)	<0.001
Malnutrition	26 (1.8)	14 (5.2)	0.003
Smoker	335 (24.7)	125 (44.2)	<0.001
Current medications, No. (%)			
Non-Steroidal Anti-Inflammatory Drugs	46 (3.4)	6 (2.3)	0.429
Angiotensin-Converting Enzyme Inhibitors	339 (22.1)	93 (29.8)	0.002
Angiotensin II Receptor Blockers	223 (14.6)	70 (22.5)	<0.001
Inhaled Corticosteroids	134 (8.7)	39 (12.6)	0.033
Systemic Corticosteroids	39 (2.5)	16 (5.2)	0.018
Vital signs at admission, median (IQR)			
Temperature °C	37 (1.2)	37.2 (1.27)	0.030
Heart Rate, Beats Per Minute	89 (21)	89 (24.2)	0.849
Oxygen Saturation in Room Air, %	95 (5)	89 (11)	<0.001
Admission signs and symptoms, No. (%)			
Fever	1197 (75.7)	225 (71.9)	0.206
Malaise	652 (41.7)	140 (45.6)	0.281
Upper Respiratory Tract Symptoms	353 (22.5)	68 (22.1)	0.879
Dyspnea	850 (54)	221 (70.4)	<0.001
Chest Pain	163 (10.4)	20 (6.4)	0.042
Cough	1070 (68)	188 (60.1)	0.015
Sputum Production	187 (11.9)	45 (14.5)	0.244
Hemoptysis	29 (1.8)	4 (1.3)	0.653
Myalgia/Arthralgia	349 (22.3)	25 (8.1)	<0.001
Headache	168 (10.7)	12 (3.9)	<0.001
Altered Consciousness	66 (4.2)	29 (9.3)	<0.001
Seizures	8 (0.5)	0 (0)	0.433
Abdominal Pain	55 (3.5)	10 (3.2)	0.937
Vomiting/Nausea	201 (12.8)	18 (5.8)	0.001
Diarrhea	290 (18.5)	37 (11.8)	0.007
Skin Rash	9 (0.6)	1 (0.3)	0.897
Laboratory findings, median (interquartile range)			
Hemoglobin, G/L	13.9 (1.9)	13 (3.1)	<0.001
White Blood Cell Count, X10^9^/L	6480 (3480)	7760 (5240)	<0.001
Lymphocyte Count - Cells/ μL	1000 (700)	800 (600)	<0.001
Neutrophil Count, Cells/ μL	4800 (3200)	6200 (4450)	<0.001
Hematocrit, %	41.5 (6)	39.2 (8.8)	<0.001
Platelets, X10^9^/L	211 (108)	195 (118)	<0.001
Activated Partial Thromboplastin Time	25.9 (3.8)	27.2 (5.1)	<0.001
International Normalized Ratio	1.08 (0.13)	1.13 (0.23)	<0.001
Aspartate Aminotransferase, U/L	36 (32)	30 (24.2)	<0.001
Alanine Aminotransferase, U/L	38 (27)	47 (36.8)	<0.001
Glucose, mg/dL	110 (34)	130 (56)	<0.001
Creatinine, mg/dL	0.96 (0.39)	1.25 (0.805)	<0.001
Sodium, mEq/L	139 (5)	139 (6)	0.8761
Potassium, mEq/L	4.3 (0.6)	4.4 (0.8)	0.029
C-Reactive Protein, mg/L	61.1 (98.5)	112 (144)	<0.001
Radiology			
Pathological chest X-ray on admission, No. (%)	1424 (91.2)	287 (91.4)	0.478

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
