# Peer review of "The PANDEMYC Score. An Easily Applicable and Interpretable Model for Predicting Mortality Associated With COVID-19"

_jcm, 2020, doi:10.3390/jcm9103066_

Round 1

Reviewer 1 Report

This a score aiming at predicting a severe outcome of COVID-19. Such studies are not unique and different score systems achieve high predictive values like this study. Important common scoring systems like the qSOFA was not included for comparison and may be just as good as the proposed system

However, such a system is not user friendly including 9 different parameters and is unlikely to ever be used.

In clinical practice some key parameters like CRP, IL-6, PaO2, D-dimer are monitored at regular intervals and are probably much better predictors when looked at over 2 or 3 days.

Author Response

This a score aiming at predicting a severe outcome of COVID-19. Such studies are not unique and different score systems achieve high predictive values like this study. Important common scoring systems like the qSOFA was not included for comparison and may be just as good as the proposed system

RESPONSE: While we are aware that other studies based on prediction models have been published, this one has some peculiarities that, in our opinion, provide useful information. Our simple model uses widely accessible clinical and laboratory data that enable us to prioritize patients, especially during a pandemic, when healthcare resources are limited. We found advanced age to be the main indicator of poor prognosis in various series of COVID-19. Other factors that have been associated with poor outcome include male gender, obesity, cancer, a high score on the sequential organ failure assessment (SOFA) scale, high concentrations of inflammatory or coagulation markers, high total and CD8 lymphocyte counts, high serum cytokine levels, and specific findings in imaging studies. We found that underlying diseases, such as hypertension, respiratory rate, impaired mental status, and other comorbidities, were independently associated with an increased risk of mortality. However, since none of these conditions improved the model's discriminatory capacity, they were eliminated.

However, such a system is not user friendly including 9 different parameters and is unlikely to ever be used.

RESPONSE: We are aware that other prognostic scores use fewer variables. Our score is based on variables that are quickly and easily collected and reproducible in any hospital at admission. The purpose of adding 9 parameters was to optimize the AUROC of the model. The parameters included age, smoking status, oxygen saturation, complete blood count, and biochemical parameters that are virtually always assessed on admission to the emergency room. In order to facilitate the calculation of the score and make it more user-friendly, we created an online tool (pandemyc-score.com) to help physicians. The score is already being used at our institution.

In clinical practice some key parameters like CRP, IL-6, PaO2, D-dimer are monitored at regular intervals and are probably much better predictors when looked at over 2 or 3 days.

RESPONSE: As commented on in the previous response, this model was designed to be used at admission. Some parameters that have been associated with a poorer response (IL-6, ferritin, D-dimer) are not routinely assessed at admission, and other parameters that help to classify the patient’s severity (eg, Pa02) are more invasive. The score could be a useful triage tool that would facilitate quick decision making when managing patients with COVID-19.

Reviewer 2 Report

In their manuscript, titled “The PANDEMYC score. An easily applicable and interpretable model for predicting mortality associated with COVID-19”, Torres-Macho et al seek to derive risk score to predict COVID-19-related inpatient mortality. They utilize a retrospective cohort of 1968 patient in a single hospital in Spain in an area highly affected by the outbreak.

One of the novelties and strong points of this paper is the utilization of the weight of evidence (WoE) transformation to input variables, allowing for measuring their relative strength in affecting class segregation and effect on the target variable. In addition to this, this work makes several interesting novel observations. The following additional questions and analyses, divided into major and minor, will need to be addressed:

Major:

  • A strength of the study is the division of the patient cohort into three discrete groups including a separate validation cohort. It is not clear if additional cross validation was used in model training and testing – the relative number of patients in each final cohort is relatively low. A further explanation of this should be added to the methods. If this was utilized, what was the effect on the AUC range? This is part of the TRIPOD criteria.
  • The authors comment on obtaining equivalent results in terms of AUC performance with both LR and gradient boosting, which helps to solidify their results. They note that ‘we compared our model with others models obtained using highly competitive ML methods’ – if additional methods were used, this information should be added as it helps to further solidify their findings.
  • Mechanical ventilation is not included in the variables tested, which would be highly relevant from a clinical perspective. Is this information available? Otherwise, this paper has very detailed information on patient variables, which is a major strength.

Minor:

  1. Table 1 lists a very large number of comparisons, utilizing either a U or chi-square test with a cutoff of 0.05. Given the potential to observe an association purely by chance, a correction for multiple comparisons should be utilized to increase stringency.
  2. The use of multiple steroid therapies were examined by the authors in this study. Please clarify if this refers to outpatient use of steroids prior to admission and does not encompass any inpatient corticosteroid therapy. This clarification should be added to the text given the recent finding of dexamethasone benefit in COVID cases.
  3. The authors used tree classification to guide variable binarization – was this compared to random forest or performed multiple times to ensure reproducibility?
  4. Although the authors’ analysis is quite robust, the overall conclusions of this paper are limited to being performed in a single center. Additionally, given that the hospital was particularly affected by COVID-19, there is a question of how much utilization of resources may have played a role in the final outcome. Thus, the inclusion of geographically separate clinical cohort would be beneficial to ensure that the authors’ findings are reproducible. This is a minor issue.
  5. Although informative, I do not have a strong preference that Table 2 be included in the final publication – it may be submitted as supplemental data or simply omitted with a comment in the text.

Author Response

A strength of the study is the division of the patient cohort into three discrete groups including a separate validation cohort. It is not clear if additional cross validation was used in model training and testing – the relative number of patients in each final cohort is relatively low. A further explanation of this should be added to the methods. If this was utilized, what was the effect on the AUC range? This is part of the TRIPOD criteria.

RESPONSE: We have included this information in the Methods section and in the Supplementary data.

The authors comment on obtaining equivalent results in terms of AUC performance with both LR and gradient boosting, which helps to solidify their results. They note that ‘we compared our model with others models obtained using highly competitive ML methods’ – if additional methods were used, this information should be added as it helps to further solidify their findings.

RESPONSE: We have included this information in the Methods section and in the Supplementary data.

Mechanical ventilation is not included in the variables tested, which would be highly relevant from a clinical perspective. Is this information available? Otherwise, this paper has very detailed information on patient variables, which is a major strength.

RESPONSE: Mechanical ventilation was not added in the model as it was considered a therapeutic approach; the model aimed to identify patient characteristics rather than interventions.

Table 1 lists a very large number of comparisons, utilizing either a U or chi-square test with a cutoff of 0.05. Given the potential to observe an association purely by chance, a correction for multiple comparisons should be utilized to increase stringency.

RESPONSE: Table 1 does not include multiple comparisons. Table 2 now includes a Bonferroni correction. We have adjusted the Methods section accordingly (lines 162-163).

The use of multiple steroid therapies were examined by the authors in this study. Please clarify if this refers to outpatient use of steroids prior to admission and does not encompass any inpatient corticosteroid therapy. This clarification should be added to the text given the recent finding of dexamethasone benefit in COVID cases.

RESPONSE: The study considers the patient’s medication at admission. Our score did not take into account the various interventions or therapies used during hospitalization.

The authors used tree classification to guide variable binarization – was this compared to random forest or performed multiple times to ensure reproducibility?

RESPONSE: We used a validation table to ensure the generalization capacity of the trees used and for purposes of binarization. Thus, the training table is used to adjust a tree to each of the explanatory variables, and the validation table is used to prune these trees Since each tree has a maximum depth of 2, the maximum possible number of leaves is 4. The process of segmentation of each tree yields 3 sub-trees: one with 2 variables, another with 3, and, finally, one with 4. This approach aimed to ensure minimum error in the validation table and to ensure that the model was logical.

Although the authors’ analysis is quite robust, the overall conclusions of this paper are limited to being performed in a single center. Additionally, given that the hospital was particularly affected by COVID-19, there is a question of how much utilization of resources may have played a role in the final outcome. Thus, the inclusion of geographically separate clinical cohort would be beneficial to ensure that the authors’ findings are reproducible. This is a minor issue.

RESPONSE: Although the study was carried out in a single centre, the sample was large (nearly 2000 patients), thus enabling us to stratify into training, validation, and test samples. While external validation was considered by the research team, it was ruled out owing to the robustness of our model.

Although informative, I do not have a strong preference that Table 2 be included in the final publication – it may be submitted as supplemental data or simply omitted with a comment in the text.

RESPONSE: We have included the table as supplementary data.

Reviewer 3 Report

Proposed paper is interesting, however some revisions are needed

  • A recent important paper on the topic has been forgotten, please cite: Hypertension. 2020 Aug;76(2):366-372.
  • It is possible that hospital saturation (has stated by the authors at the begining of the methods) affect also the mortality and it would be hopeful that in the near it is doesn't reached again. Could you check if the presence of hospital saturation during the day of admission of the patients correlate with it's outcome. If yes check if it need to be inserted into the model and if it increase the AUC.
  • Altough chest-X-Ray is most widely available many diagnoses were done more frequently on CT scan than on X-ray or swab. CT data are also needed and it's necesssity to be inserted into the score need to be checked.
  • Blood Pressure values are needed and must be inserted in tables and model.
  • Minor comment: instrction regarding paper preparation has been copy at the beginning of the introduction --> please delete it.

Author Response

A recent important paper on the topic has been forgotten, please cite: Hypertension. 2020 Aug;76(2):366-372.

RESPONSE: Thank you. We have added this reference.

It is possible that hospital saturation (has stated by the authors at the begining of the methods) affect also the mortality and it would be hopeful that in the near it is doesn't reached again. Could you check if the presence of hospital saturation during the day of admission of the patients correlate with it's outcome. If yes check if it need to be inserted into the model and if it increase the AUC.

RESPONSE: The level of saturation could have affected patient outcome, especially when ICU beds were not available. Unfortunately, we did not record saturation as a variable. However, as saturation involves many components (available ICU beds, patient-to-physician ratio, new admissions, number of transfers to other hospitals on a daily basis), we were unable to investigate it. The absence of saturation from the model has now been included as a limitation.

Altough chest-X-Ray is most widely available many diagnoses were done more frequently on CT scan than on X-ray or swab. CT data are also needed and it's necesssity to be inserted into the score need to be checked.

RESPONSE: While we agree with this comment, only 168 CT scans were performed at our institution, and the most frequent reason for the request was to rule out pulmonary thromboembolism. Therefore, we did not consider including it in the model/score.

Blood Pressure values are needed and must be inserted in tables and model.

RESPONSE: Blood pressure values (systolic blood pressure and diastolic blood pressure at admission) were added to the model; however, since neither of these values improved the model's discriminatory capacity, they were eliminated.

Minor comment: instrction regarding paper preparation has been copy at the beginning of the introduction --> please delete it.

RESPONSE: We have deleted the text as requested.

Round 2

Reviewer 3 Report

All the query raised has been answered